# Intranasal Atomization of Ketamine, Medetomidine and Butorphanol in Pet Rabbits Using a Mucosal Atomization Device

**DOI:** 10.3390/ani13132076

**Published:** 2023-06-23

**Authors:** Mitzy Mauthe von Degerfeld, Matteo Serpieri, Giuseppe Bonaffini, Chiara Ottino, Giuseppe Quaranta

**Affiliations:** Centro Animali Non Convenzionali, Dipartimento di Scienze Veterinarie, Università degli Studi di Torino, Largo Paolo Braccini 2, 10095 Grugliasco, Italy; mitzy.mauthe@unito.it (M.M.v.D.); giuseppe.bonaffini@unito.it (G.B.); chiara.ottino@unito.it (C.O.); giuseppe.quaranta@unito.it (G.Q.)

**Keywords:** mucosal atomization device, intranasal atomization, rabbit, anesthesia, ketamine, medetomidine, butorphanol, atipamezole, sedation

## Abstract

**Simple Summary:**

Intranasal atomization of drugs, using a mucosal atomization device, has gained interest in human medicine, especially in the pediatric field, as a non-invasive method compared to other parenteral routes, with fast and effective absorption of drugs. The aim of this study was to evaluate the effects of intranasal atomization, compared to intramuscular administration, of a mix of anesthetic drugs in pet rabbits. The results suggest that intranasal atomization produces anesthesia with a slightly lower depth compared to intramuscular administration, avoiding the algic stimulus related to the inoculation of drugs.

**Abstract:**

A non-invasive method of drug delivery, intranasal atomization, has shown positive results in human medicine and in some animal species. The objective of this study was to evaluate the effects of intranasal atomization, compared to intramuscular administration, of a mix of anesthetic drugs in pet rabbits. In total, 104 mixed-breed pet rabbits, undergoing various types of surgery, received a combination of ketamine, medetomidine, and butorphanol (20, 0.4, and 0.2 mg/kg) by intranasal atomization using a Mucosal Atomization Device (Group MAD) or intramuscular administration (Group IM). When required, isoflurane was dispensed through a face mask. At the end of the procedures, atipamezole was administered using the same routes in the respective Groups. There were no differences in time to loss of righting reflex between the groups, while differences were found for the need for isoflurane (higher in Group MAD) and recovery time, occurring earlier in Group MAD. The results suggest that intranasal atomization of a combination of ketamine, medetomidine, and butorphanol produces a lighter depth of anesthesia in pet rabbits, compared to intramuscular administration. Intranasal atomization can be performed to administer sedative and anesthetic drugs, avoiding the algic stimulus related to the intramuscular inoculation of drugs.

## 1. Introduction

Rabbits are increasingly popular pets, ranking third among the most common pet mammals after dogs and cats. Some medical and surgical procedures performed on rabbits require sedation or anesthesia [1,2]; however, anesthetic procedures in rabbits can represent a challenge, as perianesthetic mortality is estimated to be higher than for dogs and cats [3].

Different combinations of drugs, including ketamine, medetomidine, and butorphanol, have been used in rabbits and rodents to obtain anesthesia (drug-induced unconsciousness characterized by controlled but reversible depression of the central nervous system and perception, with the patient not arousable by noxious stimulation) or sedation (state characterized by central depression and drowsiness), partially reversible using atipamezole [4,5,6,7,8]. Furthermore, isoflurane is also commonly used for maintenance of anesthesia in rabbits [5,6].

Generally, anesthetic and sedative drugs are administered via intravenous (IV), intramuscular (IM) or subcutaneous (SC) routes [4,7]. However, the IV route is relatively impractical in conscious rabbits, as sedation is often required for the placement of an IV catheter, particularly in nervous or frightened animals [6]. The IM route is commonly used in rabbits, although violent reactions are quite frequent following the painful stimulus caused by the injection, with the risk of musculoskeletal trauma and vertebral fractures. Furthermore, depending on the drugs used and due to the relatively large drug volume often required, muscle necrosis and discomfort may occur following injection [4,6]. The SC route, generally better tolerated by rabbits, presents some disadvantages in the absorption of drugs, especially when an α2-adrenoceptor agonist (α2-AA) is used, as absorption via the SC route may be slow or erratic due to the local vasoconstrictive effect of these drugs [6,8,9,10,11].

The intranasal (IN) route, as an alternative to those previously mentioned, has recently gained attention both in human medicine, particularly in the pediatric field, and in veterinary medicine, as this route enables the avoidance of the painful stimulus related to the inoculation of drugs via other parenteral routes, and is more easily accepted by children requiring the administration of drugs for procedural sedations, premedication, or for pain management [12,13]. Another advantage of the IN route is the avoidance of the hepatic first-passage effect, allowing direct molecules passage to the central nervous system (CNS) through the olfactory epithelium, where intercellular junctions are permeable. The abundant vascularization of the olfactory mucosa guarantees an excellent absorption surface for drugs administered via the IN route [14,15,16,17]. A further passage to the CNS is allowed by the direct uptake of molecules through the trigeminal and olfactory nerve pathways [17].

On a practical level, intranasal administration can be performed by intranasal instillation of drops (IND) into the nasal cavity, or by intranasal atomization (INA) of the drugs in it [13].

In veterinary medicine, IND of sedative and anesthetic drugs has been evaluated in several species, including dogs, cats, pigs [9,18,19], and rabbits, in which different combinations of drugs were used [20,21,22,23,24,25].

While IND usually only requires the use of an IV catheter connected to a syringe, INA involves the use of a mucosal atomization device (MAD), capable of creating particles of 30–100 µm in diameter applying a high pressure on the plunger of a syringe connected to the device [12,26]. INA with the MAD, compared to IND, produces less drug loss in the oropharynx, higher drug levels in the cerebrospinal fluid, greater patient acceptability, and better sedative and analgesic effects [12,13]. In human medicine, several anesthetic and sedative drugs have been intranasally atomized using the MAD [27,28]. In veterinary medicine, the MAD has been used for the administration of anticonvulsant and sedative drugs in dogs [29,30,31,32,33]. Recent studies report the use of the MAD for the administration of a single sedative or anesthetic drug, such as medetomidine and alfaxalone, and other molecules, in rabbits [34,35,36,37]; furthermore, the maximum administrable volume using the MAD without aspiration into the trachea was evaluated in Japanese White rabbits [38].

Most of the studies regarding rabbits, however, describe experimental procedures performed on healthy New Zealand White Rabbits in laboratory settings. The evaluations carried out on these animals are not automatically applicable to pet rabbits, which present a variety of breeds and sizes, as well as potential concomitant diseases that could interfere with the evaluations [2,39]. The aim of this clinical study was to compare the effects of intranasal atomization using the MAD vs. intramuscular administration of a combination of ketamine, medetomidine, and butorphanol in pet rabbits in a clinical setting.

## 2. Materials and Methods

Ethical approval was given by the Departmental Executive Committee of the Veterinary Science Department of University of Turin (report N. 247/2022).

The study included 104 pet rabbits of various sex, age, and breed, undergoing anesthesia for various types of surgical or non-surgical procedures. Obese or cachectic subjects were excluded a priori.

The subjects were hospitalized the day before the procedures. Food and water were not removed before the anesthetic procedure. Baseline (T0) heart rate (HR) and respiratory rate (RR) were obtained by thoracic auscultation and observation of chest movements.

A combination of 20 mg/kg ketamine (Lobotor^®^, 100 mg/mL, Acme S.r.l, Corte Tegge-Cavriago, RE, Italy), 0.4 mg/kg medetomidine (Dormisan^®^, 1 mg/mL, ATI Azienda Terapeutica Veterinaria S.r.l., Milan, Italy) and 0.2 mg/kg butorphanol (Nargesic^®^, 10 mg/mL, Acme S.r.l, Corte Tegge-Cavriago, Italy), mixed in the same syringe (final volume: 0.62 mL/kg), was administered to all subjects. Rabbits were randomly divided into two groups of 52 subjects each (Group IM and Group MAD). For Group IM, the drugs were administered via the IM route into the thigh muscles, connecting a 23G needle to a syringe; the rabbit was manually restrained in sternal recumbency by a technician who pressed the rabbit firmly onto the examination table, on a non-slippery surface (e.g., a towel), pressing the back of the animal with one hand and holding the animal’s head with the other hand, covering its eyes at the same time. For Group MAD, the mixture was administered using a Mucosal Atomization Device (MAD Nasal™ Intranasal Mucosal Atomization Device MAD300, Teleflex Medical S.r.l., Varedo, MB, Italy), connected to a luer-lock syringe. Each rabbit was restrained by an operator, who held the animal vertically by supporting the forelimbs and thorax with one hand and the hindlimbs with the other hand, with the back of the animal leaning on the operator. Holding the animal’s head, the anesthetist placed the tip of the MAD in one nostril. With a quick movement of the plunger, half of the content of the syringe was administered into the nostril. About 15 s later, the other half was atomized in the other nostril, in the same way.

After the administration each rabbit was placed inside a cage, in a dark and silent room. Subsequently, time to loss of righting reflex (LRR; defined as the time from administration of the drugs to recumbency and absence of voluntary movement when turning the animal on its back), and times of loss of pedal and palpebral reflexes (respectively tested by manually pinching one digit of the hindlimb and watching for its withdrawal, and by touching the medial canthus of one eye to assess the blinking reaction of the rabbit) were recorded. At 5 min following drug administration, posture, palpebral and pedal reflexes, resistance to physical restraint, and response to fur clipping were assessed to compile a modified numerical sedation score (0–12) for rabbits [21,40] (Table 1). Each evaluation was performed by the anesthetist.

After LRR, each patient was removed from the cage and prepared for the procedure; 30 mL/kg of lukewarm fluids were subcutaneously administered in the interscapular region (1:1 mixture of lactated Ringer’s and 5% dextrose; Baxter SpA, Rome, Italy).

Each patient was placed on a heating pad and connected to a multiparameter monitoring system (Infinity Delta^®^, Dräger Italia SpA, Corsico, Italy), to measure vital parameters by electrocardiography (ECG) and pulse oximetry; the oximeter probe was placed on the phalanges of a forelimb. HR, peripheral oxygen saturation (SpO_2_), and RR were recorded 5 min (T5) after administration of anesthetic drugs and at 10 min intervals afterwards. Each animal was given 1.5 L/min of 100% oxygen through a face mask (Anesthetic face mask, S, Jørgen Kruuse A/S, Langeskov, Denmark). In case of a 20% increase in HR, isoflurane was delivered (0.5–2%; IsoFlo, Zoetis Italia srl, Rome, Italy) using a vaporizer (Vapamasta 6, Anmedic AB, Vallentuna, Sweden) through a non-rebreathing respiratory system (Bain coaxial breathing system, Intersurgical, Wokingham, UK).

When spontaneous ventilation was considered superficial, or peripheral saturation was less than 95%, or in case of apnea, manual assisted ventilation was performed at 12 breaths/minute, using a 0.5 L rebreathing bag (Jørgen Kruuse A/S, Langeskov, Denmark).

At the end of the procedures, atipamezole (Sedastop^®^, 5 mg/mL, Ecuphar Italia S.r.l., Milan, Italy) was administered at a dose of 1–2 mg/kg, based on the time from drugs administration and depth of anesthesia, as assessed by the anesthetist (considering the presence of spontaneous ventilation and evaluation of palpebral and pedal reflexes). Atipamezole was intranasally atomized using the MAD in sternal recumbency in Group MAD, and intramuscularly administered into the thigh muscles in Group IM.

After atipamezole administration, times of reappearance of the pedal and palpebral reflexes, time of head lifting (when the rabbit started raising its head), chewing time (when the rabbit started exhibiting chewing movements), and recovery time (defined as the time when spontaneous movements and righting reflex were present) were recorded.

Each animal was placed in a cage prepared with soft blankets to avoid trauma during recovery. An infrared heating lamp (InfraRed Industrial Heat Incandescent, Philips Lighting, Signify Italia S.p.A, Milan, Italy) was attached to each cage and maintained for 30 min.

The procedural time was calculated from the beginning of the procedure to its end (for surgical procedures time was recorded from the first to the last algic stimulus). The duration of anesthetic procedure was calculated from the administration of the anesthetic mixture to the administration of atipamezole. The need for manual assisted ventilation and its duration, and the need for isoflurane were noted.

### Statistical Analysis

Data management and statistical analysis were performed with Microsoft Excel (Microsoft 365, 2023; Microsoft Corp., Redmond, WA, USA) and R (version 4.2.2; R Foundation for Statistical Computing, Vienna, Austria). Continuous variables were tested for normality distribution using the Shapiro–Wilk test. Variables are reported as mean and standard deviation (SD) in case of normal distribution, and median and range in case of lack of normality. Categorical variables are reported as frequency and percentage.

Two-tailed Wilcoxon rank sum test, two-tailed Student’s *t*-test, or Fisher’s exact test were performed where applicable to evaluate homogeneity between the Groups in the whole sample for the following variables: sex, weight, breed, and age. The same tests were performed for the above-mentioned variables, with procedural time and duration of anesthetic procedure, in rabbits undergoing ovariectomy only.

Two-tailed Wilcoxon rank sum test, two-tailed Student’s *t*-test, or Fisher’s exact test were performed where applicable to compare time to LRR, times of loss of pedal and palpebral reflexes, and sedation scores between the Groups. For statistical purposes, total sedation score results were further evaluated using the 3 categories shown in Table 1 (0–3 insufficient, 4–7 moderate, 8–12 deep) and differences between Groups were analyzed with Fisher’s exact test. In rabbits undergoing ovariectomy only, the same tests were performed to compare the need for isoflurane, the need for manual assisted ventilation and its duration, atipamezole dose, physiological variables (HR, RR, and SpO_2_), and post-operative variables (times of reappearance of pedal and palpebral reflexes, chewing time, time of head lifting, and recovery time).

For rabbits undergoing ovariectomy, Friedman’s test and subsequent Wilcoxon signed rank test with Bonferroni correction were performed to evaluate HR and RR within each Group at different time points, from T0 to T30. The tests were not performed for SpO_2_, as the detection of peripheral saturation was not always possible, and several values were missing.

The rationale to form a subgroup including rabbits undergoing ovariectomy only was that some variables could have been influenced by the surgical stimulus and the duration of anesthesia. Thus, only rabbits subjected to the same kind of surgical procedure (in this case, ovariectomy was chosen) were considered to perform the statistical tests on the above-mentioned variables.

Statistical significance was set at *p* < 0.05.

## 3. Results

The distribution of breeds in the Groups is summarized in Table 2, while results regarding weight, age, time to LRR, times of loss and reappearance of the pedal and palpebral reflexes, chewing time, time of head lifting, recovery time, atipamezole dose, duration of anesthetic procedure, and procedural times are summarized in Table 3. Time to LRR and times of loss of pedal and palpebral reflexes in the Groups are represented in Figure 1. Performed procedures are listed in Table 4.

The distribution of females 22/52 (42.3%) and males 30/52 (57.7%) was the same in both Groups (*p* = 1.00). Comparison between the Groups for breed distribution was non-statistically significant (*p* = 0.176).

Following the administration of the anesthetic drugs with the MAD, some rabbits struggled for a few seconds. Numerical data regarding a behavioral response to the administration of drugs were not recorded. In total, 11 out of 52 (21.2%) rabbits sneezed after intranasal atomization, without evident leakage of drug out of the nostrils. No other complications were noted following INA.

Results regarding sedation scores are summarized in Table 5 and Table 6 and represented in Figure 2. There was a statistically significant difference for total score categories between the Groups (*p* = 0.013).

### Rabbits Undergoing Ovariectomy

The subgroup with rabbits undergoing ovariectomy included 28/104 (26.9%) subjects, of which 13/28 (46.4 %) in Group IM, and 15/28 (53.6%) in Group MAD.

There were no statistically significant differences in age, breed distribution, and weight between the Groups (*p* = 0.575, *p* = 0.607, and *p* = 0.890, respectively).

Rabbits requiring isoflurane were 0/13 (0%) in Group IM and 3/15 (20%) in Group MAD (*p* = 1.5^10^−5^).

Rabbits requiring manual assisted ventilation were 4/13 (26.7%) in Group IM and 11/15 (73.3%) in Group MAD (*p* = 0.102). The duration of manual assisted ventilation was 30 (10–45) min in Group IM, and 20 (5–40) min in Group MAD (*p* = 0.015).

In 2/15 (13.3%) cases, both in Group MAD, atipamezole was not administered as some reflexes (palpebral and/or pedal) were already present at the end of the procedures.

Differences were found in HR over time for Group MAD (*p* = 4^10^−5^) and for Group IM (*p* = 6^10^−6^). For Group MAD, differences were found between T0 and T5 (*p* = 0.007), and between T0 and T10 (*p* = 0.008). For Group IM, differences were found between T0 and T5 (*p* = 0.015), T0 and T10 (*p* = 0.015), T0 and T20 (*p* = 0.015). Results and differences between the Groups for HR at different time points are summarized in Table 7.

Differences were found in RR over time for Group MAD (*p* = 2^10^−7^) and for Group IM (*p* = 7^10^−10^). For Group MAD, differences were found between T0 and T5 (*p* = 0.009), T0 and T10 (*p* = 0.006), T0 and T20 (*p* = 0.006). For Group IM, differences were found between T0 and T5 (*p* = 0.013), T0 and T10 (*p* = 0.013), T0 and T20 (*p* = 0.012), T0 and T30 (*p* = 0.012), T5 and T10 (*p* = 0.030), T5 and T20 (*p* = 0.030), T5 and T30 (*p* = 0.030). Results and differences between the Groups for HR at different time points are summarized in Table 7.

No mortality was recorded.

## 4. Discussion

INA with a combination of ketamine, medetomidine, and butorphanol, using a mucosal atomization device, resulted in lighter anesthetic depth compared to IM administration of the same mix in pet rabbits, in accordance with preliminary results obtained on a reduced sample [41].

Few articles regarding INA with the MAD in rabbits can be found in the literature: in this section, studies regarding IND will also be considered, although the method of administration is different [12,13].

In the current study, the animals were pet rabbits of various breeds and sizes; however, the Groups can be considered homogeneous as no significant differences were found in weight, age, sex, and breed.

The use of the IN route has determined the need to identify the optimal position of the animal to ease administration. In the studies of Wei et al. [36,37,38] the rabbits were restrained in sternal recumbency, and no data regarding any complications related to INA with the MAD in this position were reported. In the study by Santangelo et al. [21], rabbits were held in a “sitting” position on the edge of a table, and the Authors noted swallowing attempts by some animals during IND. In the study of Weiland et al. [25], two rabbits experienced epistaxis after IND in dorsal recumbency. In the present study, 11/52 (21.2%) sneezed after INA of the anesthetic mixture, as reported in dogs [9,33]; also, Yanmaz et al. [24], recorded sneezing in 25% of cases after IND in rabbits. The position used in our study facilitated effective restraint in the rabbits during INA. The animals included in the study were pet rabbits, therefore probably accustomed to light manual restraint by their owners. The administration of atipamezole was performed in sternal recumbency in relaxed and unconscious rabbits, and no complications were noted.

The volume of drug used in our study (0.62 mL/kg), divided between the nostrils, is higher than that recommended for the use of the MAD by Wei et al. [38] to avoid leakage from the nasal cavity towards the nasopharynx or trachea. However, the volume used in our study is comparable to that employed by Yanmaz et al. [24] (0.60 mL/kg) for IND of midazolam and dexmedetomidine in rabbits.

No significant differences were found comparing time to LRR and times of loss of palpebral and pedal reflexes between the Groups, although a greater variability was found in Group MAD for the mentioned variables, as shown by the interquartile ranges in Figure 1. A previous study reported lower times to LRR after IND of midazolam and dexmedetomidine in rabbits, compared to IM administration. This divergence from our study could be explained by the sternal position used by Yanmaz et al. [24], which could have positively influenced the absorption of the mixture. Similarly, IND of a mix of midazolam and sufentanil resulted in a faster LRR compared to IM administration in eleven rabbits in another study [23]. Times to LRR in our study are comparable to those reported for the same mixture administered IM, at slightly different dosages [4].

Also, based on a modified numerical scale for rabbits [31,40], differences were found in spontaneous posture, palpebral and pedal reflex, and total score between the Groups, with deep sedation more frequently obtained in Group IM (Figure 2). These results differ from those reported by Micieli et al. [31] in dogs, where the sedation score following INA of dexmedetomidine was higher than after intramuscular administration; our results are, instead, comparable with those obtained on dogs by other Authors [32,33], in which INA of medetomidine and dexmedetomidine, resulted in lower sedation scores compared to IM administration.

The duration of anesthetic procedure and procedural time did not show significant differences for rabbits undergoing ovariectomy, denoting homogeneity between the Groups.

The significant difference found for the need for isoflurane, more frequent in Group MAD, suggests a lower depth of anesthesia following INA; on the other hand, rabbits in Group IM had a more frequent need for manual assisted ventilation, denoting a greater depressant effect on the respiratory system using this route.

The HR decreased over time in both Groups, particularly at T5, probably due to the negative chronotropic effect of the α2-AA [42]. However, median HR was higher after INA than after IM administration (Table 7), similar to that reported in the literature for dogs [31,33]. This finding could be due to a direct local vasoconstrictive effect caused by medetomidine on the nasal mucosa, resulting in slower or erratic absorption of this drug and less related cardiovascular effects in Group MAD, also possibly explaining variability in LRR and differences in the sedation score. In addition to previous assumptions regarding HR, Micieli et al. [31], hypothesized a possible greater direct passage to the CNS following INA of dexmedetomidine in dogs, with less systemic vasoconstrictive effect, usually causing consequent transient hypertension and reflex bradycardia. The absence of blood pressure monitoring in our study represents an important limitation for the evaluation of the related effects following the administration of drugs. The need for isoflurane, higher in Group MAD, could have also influenced HR for the vasodilatative and hypotensive effects of the volatile agent, which generally induces a compensatory increase in heart rate [43].

Also, the median RR decreased over time in both groups, probably due to the depressant effect on the respiratory system caused by the drugs contained in the mixture. In our case, the differences are particularly noticeable from T10 onwards. This discrepancy is certainly influenced by the manual assisted ventilation required in many cases in Group IM, set at 12 breaths per minute. This represents an important limitation for the evaluation of the effects on the respiratory rate according to the route of administration.

With regard to SpO_2_, it was not possible to report the median and range at T5 in Group IM, as at that time the parameter was not detectable in rabbits undergoing ovariectomy. This could be due to the peripheral vasoconstrictive effect of medetomidine, presumably greater in Group IM, preventing the measurement of SpO_2_ in this Group at T5 [44]. The progressive resolution of the vasoconstrictive effect and possible vasodilation induced by isoflurane could therefore have made the SpO_2_ detectable from T10. All animals received 100% oxygen and, in both Groups, median SpO_2_ was above 95% in all measurements, a value usually considered not indicative of hypoxemia [45].

Following the administration of atipamezole, statistically significant differences were found in times of reappearance of palpebral and pedal reflexes, chewing time, time of head lifting and recovery time between the Groups: all variables were lower in Group MAD, denoting faster recovery after INA. Considering the faster effects in Group MAD, with lower medians compared to Group IM (Table 3), it could be hypothesized that the antagonist, when intranasally administered, had a faster onset than after IM administration. Unlike medetomidine, the administration of atipamezole may result in vasodilation [42]. It is possible that this effect was locally exerted in the nasal mucosa, resulting in faster absorption of the drug. Furthermore, the direct passage to the CNS, as described following INA [17], could have hastened the central effect of the α2-adrenoceptor antagonist. The lower median recovery time in Group MAD could be due to the existing lighter anesthetic depth at the end of the surgical procedures. In two cases of Group MAD, the antagonist was not required as the reflexes began to reappear. Further studies are needed to compare the use of different routes (i.e., INA vs. IM) for the administration of atipamezole after the use of a single route of α2-AA administration.

## 5. Conclusions

INA of a combination of ketamine, medetomidine, and butorphanol (20, 0.4, and 0.2 mg/kg, respectively) using a mucosal atomization device resulted in a lighter anesthetic depth compared with IM administration in pet rabbits in a clinical setting. No differences in time to LRR were found, while a median recovery time was faster after INA. Most importantly, by the use of INA, the algic stimulus related to the intramuscular inoculation of drugs is avoided.

## Figures and Tables

**Figure 1 animals-13-02076-f001:**
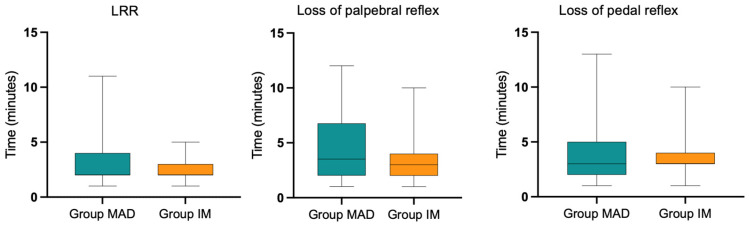
Box-plot graphs for time to LRR and times of loss of palpebral and pedal reflexes in the Groups. Boxes indicate interquartile range and median (black line). Whiskers represent minimum and maximum.

**Figure 2 animals-13-02076-f002:**
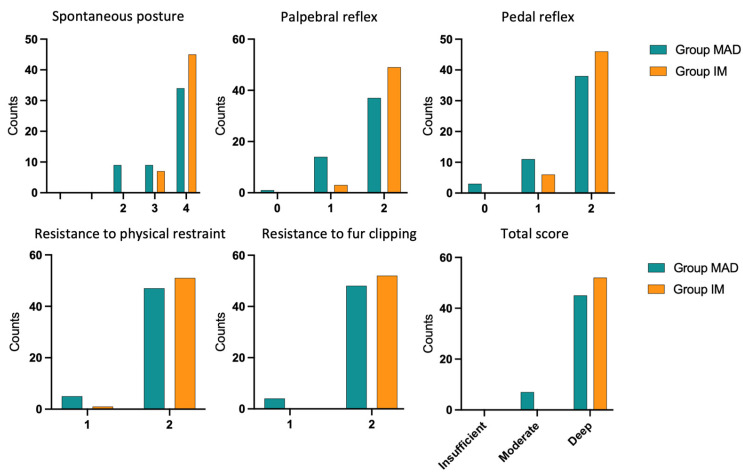
Histograms representing the sedation scores in the Groups. Total score is represented as insufficient (0–3), moderate (4–7), deep (8–12).

**Table 1 animals-13-02076-t001:** Modified numerical sedation score for rabbits [21,40].

Parameter	Behavior of the Rabbit	Score
Spontaneous posture	Normal	0
	Lying sternally, head up	1
	Lying sternally or laterally, responding to stimuli	2
	Lying, not responding to stimuli	3
	Complete muscle relaxation	4
Palpebral reflex	Normal	0
	Decreased	1
	Absent	2
Response to fur clipping	Normal response	0
	Reduced	1
	Absent	2
Resistance to physical restraint	Normal resistance	0
	Moderate resistance	1
	No resistance	2
Pedal reflex	Normal	0
	Decreased	1
	No reaction	2
Total	Insufficient	0–3
	Moderate	4–7
	Deep	8–12

**Table 2 animals-13-02076-t002:** Descriptive analysis (number (%)) of the breeds in the Groups.

Breed	Group IM	Group MAD
Angora	-	1 (1.9%)
Dutch	1 (1.9%)	-
Dwarf	11 (21.2%)	12 (23.1%)
Dwarf Lop	26 (50.0%)	26 (50.0%)
Lionhead	13 (25.0%)	7 (13.5%)
New Zealand White	-	2 (3.8%)
Polish	2 (3.8%)	3 (5.8%)

**Table 3 animals-13-02076-t003:** Descriptive analysis (median (range) or mean ± SD) and *p* values of weight, age, time to LRR, times of loss and reappearance of palpebral and pedal reflexes, chewing time, time of head lifting, recovery time, atipamezole dose, duration of anesthetic procedure, and procedural time in Groups IM and MAD.

Variable	Group IM	Group MAD	*p* Value
Weight (kg)	1.75 (1.33–3.00)	1.70 (1.12–3.90)	0.971
Age (months)	8 (5–30)	8 (5–36)	0.139
Time to LRR (min)	2 (1–5)	2 (1–11)	0.709
Loss of palpebral reflex (min)	3 (1–10)	3 (1–12)	0.396
Loss of pedal reflex (min)	3 (1–10)	3 (1–13)	0.669
Reappearance of palpebral reflex (min) ^a^	3 (1–10)	1 (1–3)	0.026 *
Reappearance of pedal reflex (min) ^a^	5 (2–12)	1 (1–3)	1^10^−4^ *
Chewing time (min) ^a^	6 (1–17)	3 (1–6)	0.002 *
Time of head lifting (min) ^a^	7 (3–16)	2 (1–6)	9^10^−5^ *
Recovery time (min) ^a^	11 (4–20)	4 (1–21)	0.002 *
Atipamezole dose (mg/kg) ^a^	1.94 (1.66–2.06)	1.00 (0.00–2.00)	0.046 *
Duration of anesthetic procedure (min) ^a^	41.1 ± 8.1	45.6 ± 10.7	0.199
Procedural time (min) ^a^	25.8 ± 6.2	30.5 ± 9.9	0.135

^a^: Analysis performed on data from rabbits undergoing ovariectomy. *: statistically significant differences between the Groups.

**Table 4 animals-13-02076-t004:** Descriptive analysis (number (%)) of the procedures performed in the Groups.

Procedure	Group IM	Group MAD
Abscess marsupialization	2 (3.8%)	-
Cystoscopy	-	1 (1.9%)
Cystotomy	1 (1.9%)	1 (1.9%)
Caesarean section	1 (1.9%)	-
Enucleation	2 (3.8%)	-
Oral endoscopy	-	2 (3.8%)
Orchiectomy	9 (17.3%)	19 (36.5%)
Osteosynthesis	3 (5.8%)	3 (5.8%)
Ovariectomy	13 (25.0%)	15 (28.8%)
Ovariohysterectomy	2 (3.8%)	2 (3.8%)
Teeth extraction	14 (26.9%)	5 (9.6%)
Teeth trimming	3 (5.8%)	4 (7.7%)

**Table 5 animals-13-02076-t005:** Comparison of the sedation score between the Groups.

	Group IM	Group MAD	
Score (Min–Max)	Median (Range)	Median (Range)	*p* Value
Spontaneous posture (0–4)	4 (3–4)	4 (2–4)	0.006 *
Palpebral reflex (0–2)	2 (1–2)	2 (0–2)	0.002 *
Pedal reflex (0–2)	2 (1–2)	2 (0–2)	0.039 *
Resistance to physical restraint (0–2)	2 (1–2)	2 (1–2)	0.096
Response to fur clipping (0–2)	2 (2–2)	2 (1–2)	0.390
Total (0–12)	12 (9–12)	12 (6–12)	0.005 *

*: Statistically significant differences between the Groups.

**Table 6 animals-13-02076-t006:** Total sedation score categories, using a modified scale for rabbits [19,38], in the Groups.

Total Score	Group IM	Group MAD
Insufficient (0–3)	0 (0%)	0 (0%)
Moderate (4–7)	0 (0%)	7 (13.5%)
Deep (8–12)	52 (100%)	45 (86.5%)

**Table 7 animals-13-02076-t007:** Analysis of the physiological variables: heart rate (HR, beats/min), respiratory rate (RR, breaths/min), and peripheral saturation (SpO_2_, %), reported as median and range.

Time from Administration (Min)					
Variable	T0	T5	T10	T20	T30
HR					
Group IM	208 (170–280)	120 (90–181)	127 (97–176)	132 (90–168)	140 (96–197)
Group MAD	208 (134–280)	140 (97–200)	140 (98–227)	155 (95–220)	177 (131–235)
*p* value	0.745	0.031 *	0.406	0.002 *	0.013 *
RR					
Group IM	120 (80–168)	32 (12–100)	12 (12–24)	12 (12–12)	12 (12–12)
Group MAD	120 (80–120)	40 (12–120)	28 (12–60)	12 (12–36)	12 (12–60)
*p* value	0.564	0.945	6^10^−4^ *	0.006 *	0.006 *
SpO2%					
Group IM	-	- ^a^	99 (99–99)	100 (92–100)	96 (76–100)
Group MAD	-	98.5 (96–100)	98.5 (71–100)	100 (95–100)	100 (95–100)

^a^: Non-measurable data. *: statistically significant differences between the Groups.

## Data Availability

Further specific data regarding each animal can be requested from the Authors.

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
