# Peer review of "Intranasal Atomization of Ketamine, Medetomidine and Butorphanol in Pet Rabbits Using a Mucosal Atomization Device"

_animals, 2023, doi:10.3390/ani13132076_

Round 1
Reviewer 1 Report
The study is very interesting and gives useful information for the management of rabbit anesthesia. However it needs to be significantly improved.
Line56: Add and comment the following bibliographic note as it explains why alpha 2 agonists and butorphanol are associated: Claudia, Interlandi, Gioacchino, Calapai, Bernadette, Nastasi, Carmen, Mannucci, Manuel, Morici, Giovanna L., Costa (2017). Effects of Atipamezole on the Analgesic Activity of Butorphanol in Rats. JOURNAL OF EXOTIC PET MEDICINE, vol. 26, p. 290-293, ISSN: 1557-5063, doi: DOI: http://dx.doi.org/10.1053/j.jepm.2017.07.001
Statistical Analysis:
perform normality analysis using the Shapiro-Wilk test and report as mean or median (range) SD as appropriate. Demographics will need to be analysed using the two-tailed t-test, the two-tailed Mann-Whitney test, or Fisher's exact t. Observer agreement for sedation quality using Kendall's coefficient of agreement W. should be performed for each time point measured. Evaluate differences within treatment using the Wilcoxon test. Compare treatments with the Mann-Whitney test. Report the actual values ​​of p. Sum the sedation scores at the various time points and report them with median and range as in the cited bibliographic note. A great perplexity is the difference in the surgical stimulus, due to the different surgeries for which I advise the authors to choose only one surgery at least for the evaluation of the physiological variables. Also mention the intended rescue analgesic.Rewrite the results in more detail.
Minor editing of English language required
Author Response
Response to Reviewer 1 Comments
Thank you for your review and your suggestions. We have made the changes you suggested.
Added or modified words or sentences are highlighted in green.
Point 1: Line56: Add and comment the following bibliographic note as it explains why alpha 2 agonists and butorphanol are associated: Claudia, Interlandi, Gioacchino, Calapai, Bernadette, Nastasi, Carmen, Mannucci, Manuel, Morici, Giovanna L., Costa (2017). Effects of Atipamezole on the Analgesic Activity of Butorphanol in Rats. JOURNAL OF EXOTIC PET MEDICINE, vol. 26, p. 290-293, ISSN: 1557-5063, doi: DOI: http://dx.doi.org/10.1053/j.jepm.2017.07.001
Response 1: Thank for the suggestion. Reference added.
Point 2: Statistical Analysis:
perform normality analysis using the Shapiro-Wilk test and report as mean or median (range) SD as appropriate. Demographics will need to be analysed using the two-tailed t-test, the two-tailed Mann-Whitney test, or Fisher's exact t. Observer agreement for sedation quality using Kendall's coefficient of agreement W. should be performed for each time point measured. Evaluate differences within treatment using the Wilcoxon test. Compare treatments with the Mann-Whitney test. Report the actual values ​​of p. Sum the sedation scores at the various time points and report them with median and range as in the cited bibliographic note. A great perplexity is the difference in the surgical stimulus, due to the different surgeries for which I advise the authors to choose only one surgery at least for the evaluation of the physiological variables. Also mention the intended rescue analgesic.
Response 2: thank you for the wise suggestion. We have drastically corrected the statistical analysis section and changed the results and related discussion. The evaluation for the observer agreement could not be performed as the evaluation for the sedation score was always performed by the same operator, as now specified in the text. The sedation scores were assessed at T5 only, so it wasn’t possibile to sum the scores at various time points. We did not mention the intended rescue analgesia as in this kind of protocol, extensively used in our clinical setting, it is usually sufficient to use isoflurane to have a deeper plane of anesthesia, and it is not necessary to use rescue analgesia. In case of need, we use to add 0.05 mg/kg butorphanol IV o IM, as we use this drug during the induction phase. We did not mention this as this was not the focus of the manuscript.
Point 3: Rewrite the results in more detail.
Response 3: done.
Reviewer 2 Report
I am rejecting this manuscript due to the many grammatical errors and confusion with the abbreviations and outcome. The presentation is not scientific. It is taking me a long time to review each problem. The study is an important one and it should be published but the manuscript will have to be re-written by someone knowledgeable in the area of scientific manuscript writing, I am not going to do this. Attached is the manuscript with highlights in areas that highlight my concerns. Upon reading this manuscript you will understand my review comments.
Author Response
Response to Reviewer 2 Comments
Thank you for your review and your suggestions. We have made several changes in the text.
Added or modified words or sentences are highlighted in green.
Point 1: I am rejecting this manuscript due to the many grammatical errors and confusion with the abbreviations and outcome. The presentation is not scientific. It is taking me a long time to review each problem. The study is an important one and it should be published but the manuscript will have to be re-written by someone knowledgeable in the area of scientific manuscript writing, I am not going to do this. Attached is the manuscript with highlights in areas that highlight my concerns. Upon reading this manuscript you will understand my review comments.
Response 1: Dear Reviewer, we are very sorry that our manuscript was not adequate for publication. It is possible that the absence of an English native speaker in our team prevented us to obtain the desired result in the exposition of the concepts. We have made several changes to improve the text, also based on the suggestions made by other Reviewers. Based on the highlighted parts only, it is possible that some points to be corrected were not obvious to us and we may not have understood the problem. We hope that the corrected text will be more easily readable and can be published.
Reviewer 3 Report
Dear Authors,
many thanks for submitting this manuscript. I would have some suggestions for improvement.
Introduction :
The introduction is too long and could be shortened without loss of information. This would allow inclusion of relevant points e.g Information on drugs used and the difference in onset, duration of action, pharmacology, pH, sedation versus general anaesthesia, response to nasal pH.
There needs to be a clear definition of sedation versus reduced motor response versus general anaesthesia.
Some point would benefit from references.
Material and methods:
Induction time needs to be defined. Loss of righting reflex is not synonymous to onset of general anaesthesia.
The same applies to other variables: recovery time, duration of anaesthesia.
The main aim of the technique described would be reduced stress and discomfort on administration of drugs. There is no mention how this evaluated.
It would be beneficial to include more details on the sedation score than a reference to a different paper.
Please give more details on the assessment of induction time etc. were they assessed in 5 minutes intervals as indicated?
How was depth of anaesthesia defined? There seems to be overlap with sedation.
Duration of anaesthesia is not clearly defined. Time of administration of drugs does not equal onset of effect.
Please explain rational to form subgroup with neuters only.
Results:
please include more information on response to administration of drugs. There are no numbers, description or scores.
What is the sedation score maximum?
Please define HR and RR trend as this is not a conventional variable. Consider using HR and RR as absolute measure not trend.
Please give information on how many animals in group MAD needed isoflurane and ventilation.
Why were rabbits receiving no atipamezol removed from analysis?
Cystoscopy/cystotomy
Discussion:
Throughout the manuscript there is no distinction between results with p<0.05 and above. Please revise.
Trend, close to significance etc. are not appropriate terms when discussing results.
Please explain how route of administration could affect metabolism.
The manuscript would benefit from review by an English speaker.
Author Response
Response to Reviewer 3 Comments
Thank you for your review and your suggestions. We have made the changes you suggested.
Added or modified words or sentences are highlighted in green.
Point 1: The introduction is too long and could be shortened without loss of information. This would allow inclusion of relevant points e.g Information on drugs used and the difference in onset, duration of action, pharmacology, pH, sedation versus general anaesthesia, response to nasal pH.
Response 1: Thank you for your suggestion. We’ve shortened and removed some sentences and add some points. We focused on the accurate description of the route of administration, without analysing deeply the pharmacological characteristics of the drugs, as this study primarily aimed at comparing the effects of drug administration from a clinical point of view, and not on the pharmacokinetics or pharmacodynamics (in fact, in our work we did not measure the pH of the final mixture).
Point 2: There needs to be a clear definition of sedation versus reduced motor response versus general anaesthesia.
Response 2: definitions added.
Point 3: Some point would benefit from references.
Response 3: some references have been added.
Point 4: Induction time needs to be defined. Loss of righting reflex is not synonymous to onset of general anaesthesia.
The same applies to other variables: recovery time, duration of anaesthesia.
Response 4: thank you for your suggestions. We’ve re-defined times in the text.
Point 5: The main aim of the technique described would be reduced stress and discomfort on administration of drugs. There is no mention how this evaluated.
Response 5: We’ve corrected the simple summary, as the evaluation of stress has not been performed. Thank you.
Point 6: It would be beneficial to include more details on the sedation score than a reference to a different paper.
Response 6: done.
Point 7: Please give more details on the assessment of induction time etc. were they assessed in 5 minutes intervals as indicated?
Response 7: yes, the assessments were made as indicated. Some corrections have been made in the text.
Point 8: How was depth of anaesthesia defined? There seems to be overlap with sedation.
Response 8: explanations have been added in the text.
Point 9: Duration of anaesthesia is not clearly defined. Time of administration of drugs does not equal onset of effect.
Response 9: thank you. The duration of anesthesia has been re-defined.
Point 10: Please explain rational to form subgroup with neuters only.
Response 10: rationale added.
Point 11: please include more information on response to administration of drugs. There are no numbers, description or scores.
Response 11: thank you for highlighting this limitation. Information has been added at lines 217-220. Behavioral response has not been evaluated using scores.
Point 12: What is the sedation score maximum?
Response 12: the range has been added in Table 1.
Point 13: Please define HR and RR trend as this is not a conventional variable. Consider using HR and RR as absolute measure not trend.
Response 13: thank you for your suggestion. HR and RR have been considered as absolute measure for comparisons between Groups. The parameters were also considered “over time” for comparison at different time points within Groups. The word “trend” has been removed. The statistical analysis section has been reviewed also based on suggestions by other Reviewers.
Point 14: Please give information on how many animals in group MAD needed isoflurane and ventilation.
Response 14: the Results section has been extensively revised, including this information.
Point 15: Why were rabbits receiving no atipamezol removed from analysis?
Response 15: this has been revised and those rabbits were considered.
Point 16: Cystoscopy/cystotomy
Response 16: corrected.
Point 17: Throughout the manuscript there is no distinction between results with p<0.05 and above. Please revise.
Response 17: some corrections have been made, and significant results have been discussed.
Point 18: Trend, close to significance etc. are not appropriate terms when discussing results.
Response 18: corrected.
Point 19: Please explain how route of administration could affect metabolism.
Response 19: some hypothesis have been made in the Discussion section about how the route could affect absorption of the drugs, also citing other studies. To better explain how metabolism could be affected, pharmacokinetics and dynamics should be evaluated in futher studies.
Round 2
Reviewer 1 Report
Dear author, thank you for the opportunity to review this study which I find very interesting. Congratulations on the statistical analysis. I am pleased to have helped to make this study eligible for publication in this journal.
Author Response
Dear Reviewer,
Thank you for appreciating and for having improved our work. We have still performed some minor corrections as suggested by other reviewers. The new corrections are highlighted in yellow.
Reviewer 2 Report
The study is very good and important.
I am not a statistician so have not commented on methods and results in this section.

Still major problems with grammar. In some instances difficult to understand. Manuscript and grammatical corrections are attached.
Author Response
Dear Reviewer,
Thank you for the reviewing our work and improving it. The corrections are highlighted in yellow.
51 used drugs – change to – drugs used
Corrected.
58 human medicine - particularly in the pediatric field – and remove dash replace with commas
Corrected.
64 permeable: the remove colon replace with full stop. Permeable.
Corrected.
83-84 CORRECTED sedative drugs in dogs[29-33]. Recent studies reporting the use of the MAD for the administration of a single sedative or anesthetic drug, such as medetomidine and alfaxalone,
Corrected.
89 in laboratory settings. The evaluations
Corrected.
90 which instead remove this
Corrected.
91 disease which remove and replace with that
Corrected.
95 environment remove and replace with setting
Corrected.
109 administered to all subjects. Rabbits
Corrected.
113 (e.g., a towel),
Corrected.
117 syringe. Each
Corrected.
121 nostril. With
Corrected.
124 room. Subsequently
Corrected.
130 At 5 minutes following drug administration,
Corrected.
134 Table 1 The parameters are no aligned with behaviour and score, very confusing
We added some lines to make it clearer. We used the Animals’ Microsoft Word template for table, so we think that this issue will be solved in the final editing phase.
143 recorded 5 minutes (T5) after administration of anesthetic drugs and at 10 minute intervals for 30 minutes.
Corrected.
145 Denmark). In
Corrected.
154 -155 was administered at a dose of 1-2 mg/kg, based on time from drug administration and depth of anesthesia as assessed by the anesthetist
Corrected.
157 REMOVE , using lower dose in case of lighter plane of anesthesia. This is noted in the statement above
Corrected.
163 Recorded. REMOVE after the administration of atipamezole as this is previously stated.
Corrected.
167 was attached to each cage and maintained on for 30 minutes.
Corrected.
169 The procedural time was calculated from the beginning to its end, for surgical procedures time was recorded from the first to the last algic stimulus,
Corrected.
170 REMOVE while . The duration of anesthetic
Corrected.
177- 179. This sentence is unclear due to erroneous punctuation. Please correct.Continuous variables were tested for normality distribution using the Shapiro-Wilk test and are reported as mean and standard deviation (SD) in case of normal distribution, median and range in case of lack of normality. Categorical variables are reported as frequency and percentage
Corrected.
182 REMOVE as appropriate replace with where applicable
Corrected.
184 REMOVE along with
Corrected.
188 REMOVE in the whole sample.
Corrected.
189 furtherly REMOVE
Corrected.
191-194 The same tests were performed to compare need for isoflurane, need for manual assisted ventilation and its duration, atipamezole dose, physiological variables (HR, RR, and SpO2), and post- operative variables (times of reappearance of pedal and palpebral reflexes, chewing time, time of head lifting, recovery time) in rabbits undergoing ovariectomy only. – change to –
In rabbits undergoing ovariectomy only, the same tests were performed to compare the need for isoflurane, for manual assisted ventilation and its duration, atipamezole dose, the physiological variables (HR, RR, and SpO2) and post-operative variables (times of reappearance of pedal and palpebral reflexes, chewing time, time of head lifting, recovery time).
Corrected.
201 -204 THIS PARAGRAPH IS NOT CLEAR.The rationale to form a subgroup including rabbits undergoing ovariectomy only was that the mentioned variables could have been influenced by the surgical stimulus and the duration of anesthesia, depending on the surgical procedure: this would have introduced a bias into the statistical analysis. Statistical significance was set at p<0.05.
Corrected. We changed the sentences to make them clearer, also according to what suggested by our statistician.
210 operative times are
Corrected.
213 Rewrite: The distribution of females 22/52(42.3%) and males 30/52 (57.7%) was the same in both Groups
Corrected.
218 REMOVE and cannot be therefore reported.
Corrected.
222 There was a statistically
Corrected.
234-235 In 2/15 (13.3%) cases, both in Group MAD, atipamezole was not administered as some reflexes (palpebral and/or pedal)were already present at the end of the procedure
Corrected.
237-248. This results section is difficult to interpret. Please seek help from a statistician for statistical analysis rer-write
We sought help from the statistician of our institute. In her opinion, results in this section are well described, and she believes that a table or a graph would not drastically facilitate the illustration of those selected results.
Table 6 not aligned
With our version of Microsoft word, Table 6 is aligned. We believes that this is just a layout error of the conversion of the file from Word to Pdf, and that will be solved in the editing phase.
279 the route of administration is different [12,13].Should read method of administration. The route is still intranasal, the methods atomization vs drops is method
Corrected.
281 In the evaluated sample,- change to – in the current study
Corrected.
293 allowed to effectively restrain in the rabbits during INA; the - change to - facilitated effective restraint in the rabbits during INA. The
Corrected.
297 The volume of drug used in our case (0.62 ml/kg), although divided between the nostrils, is higher than that recommended for use of the MAD by Wei et al. [38] – change to - The volume of drug used in our study (0.62 ml/kg), divided between the nostrils, is higher than that recommended for use of the MAD by Wei et al. [38] QUESTION: Did Wei et al divide the volume?
Corrected. Wei et al administered different volumes in each nostril, and concluded that the safe volume is 0,3 ml per nostril in Japanese White Rabbits. The rabbits in the study weighed 2.99-4.28 kg, so the volume is about 0,1 ml/kg per nostril, higher than that used in the current study (0,31 ml/kg/nostril).
306 administration: this – change to – administration. This
Corrected.
323 The significant diff erence found for the need for isoflurane, more frequent Group MAD, suggests a lower depth of anesthesia following INA; on the other hand, rabbits in
Corrected.
327 probably due to the negative chronotropic effect
Corrected.
329 similar to that reported in the literature for dogs
Corrected.
348 As regards SpO – change to – With regard to
Corrected.
363 ]. It is possible
Corrected.
366-367 The lower median recovery time and reappearance of reflexes which could be due to the existing lighter anesthetic depth at the end of the surgical procedures, the antagonist was not required.
Corrected.
368-370 Further studies are needed to compare the use of different routes (i.e., INA vs IM) for the administration of atipamezole after a single route of α2-AA administration.
Corrected.
375 l setting.
Corrected.
377 Most importantly, by the use of INA any painful stimulus related to the inoculation of drugs is avoided.
Corrected.
Reviewer 3 Report
Many thanks for your reply to my comments and the changes you have made to the manuscript.
The manuscript is now well balanced and structured with the results and discussion much clearer and easy to follow.
The only major change I have to suggest is to remove the last sentence of the conclusion. The study did not assess pain on administration and there is no conclusive data presented in the manuscript supporting this statement which should therefore be removed.
Minor changes:
Line 154: remove “with”
Author Response
Dear Reviewer,
Thank you for having improved our work. We have performed some corrections as suggested by other reviewers. The corrections are highlighted in yellow.
Point 1. The only major change I have to suggest is to remove the last sentence of the conclusion. The study did not assess pain on administration and there is no conclusive data presented in the manuscript supporting this statement which should therefore be removed.
Response 1. We corrected the sentence, as also suggested by another Reviewer. The sentence is now more clearly stating that the avoided algic stimulus is that related to the intramuscular inoculation, even if the possible discomfort related to intranasal administration was not evaluated.
Point 2. Line 154: remove “with”
Response 2. Corrected.